# The [^18^F]F-FDG PET/CT Radiomics Classifier of Histologic Subtypes and Anatomical Disease Origins across Various Malignancies: A Proof-of-Principle Study

**DOI:** 10.3390/cancers16101873

**Published:** 2024-05-15

**Authors:** Ricarda Hinzpeter, Seyed Ali Mirshahvalad, Vanessa Murad, Lisa Avery, Roshini Kulanthaivelu, Andres Kohan, Claudia Ortega, Elena Elimova, Jonathan Yeung, Andrew Hope, Ur Metser, Patrick Veit-Haibach

**Affiliations:** 1University Medical Imaging Toronto, Joint Department Medical Imaging, University Health Network, Sinai Health System, Women’s College Hospital, University of Toronto, Toronto, ON M5G 2N2, Canada; ali.mirshahvalad@uhn.ca (S.A.M.); vanessa.murad@uhn.ca (V.M.); roshini.kulanthaivelu@queensu.ca (R.K.); andres.kohan@uhn.ca (A.K.); caludia.ortega@uhn.ca (C.O.); ur.metser@uhn.ca (U.M.); patrick.veit-haibach@uhn.ca (P.V.-H.); 2Institute for Diagnostic and Interventional Radiology, University Hospital Zurich, 8091 Zurich, Switzerland; 3Department of Biostatistics, Princess Margaret Cancer Centre, University Health Network, University of Toronto, Toronto, ON M5G 1X6, Canada; lisa.avery@uhn.ca; 4Division of Biostatistics, Dalla Lana School of Public Health, University of Toronto, Toronto, ON M5T 3M7, Canada; 5Department of Medical Oncology, Princess Margaret Cancer Centre, University Health Network, University of Toronto, Toronto, ON M5G 2C4, Canada; elena.elimova@uhn.ca; 6Division of Thoracic Surgery, Department of Surgery, Toronto General Hospital, University Health Network, University of Toronto, Toronto, ON M5G 2C4, Canada; jonathan.yeung@uhn.ca; 7Department of Radiation Oncology, University Health Network, Toronto, ON M5G 2C4, Canada; andrew.hope@uhn.ca

**Keywords:** fluorodeoxyglucose, PET/CT, radiomics, lung, gastroesophageal, head and neck, histology

## Abstract

**Simple Summary:**

There is a lack of ample knowledge about the origin of anatomical disease- and histology-specific radiomic variations across different malignancies. Thus, the aim of our proof-of-principle study was to investigate whether radiomics features have the ability to discriminate between histological subtypes and to predict the anatomical disease origin of different tumour entities. Also, we tried to synchronously predict histology and anatomical disease origin using baseline [^18^F]F-FDG PET/CT. Based on our findings, our proof-of-principle study may demonstrate a potentially high degree of diagnostic accuracy to predict histology and disease origin across different tumour entities, using the standard of care combined [^18^F]F-FDG PET/CT-derived radiomics features. This may help with the development of a new paradigm regarding the potential of also thinking about tumours from their molecular basis, and not only on the basis of their anatomical origin perse.

**Abstract:**

We aimed to investigate whether [^18^F]F-FDG-PET/CT-derived radiomics can classify histologic subtypes and determine the anatomical origin of various malignancies. In this IRB-approved retrospective study, 391 patients (age = 66.7 ± 11.2) with pulmonary (n = 142), gastroesophageal (n = 128) and head and neck (n = 121) malignancies were included. Image segmentation and feature extraction were performed semi-automatically. Two models (all possible subset regression [APS] and recursive partitioning) were employed to predict histology (squamous cell carcinoma [SCC; n = 219] vs. adenocarcinoma [AC; n = 172]), the anatomical origin, and histology plus anatomical origin. The recursive partitioning algorithm outperformed APS to determine histology (sensitivity 0.90 vs. 0.73; specificity 0.77 vs. 0.65). The recursive partitioning algorithm also revealed good predictive ability regarding anatomical origin. Particularly, pulmonary malignancies were identified with high accuracy (sensitivity 0.93; specificity 0.98). Finally, a model for the synchronous prediction of histology and anatomical disease origin resulted in high accuracy in determining gastroesophageal AC (sensitivity 0.88; specificity 0.92), pulmonary AC (sensitivity 0.89; specificity 0.88) and head and neck SCC (sensitivity 0.91; specificity 0.92). Adding PET-features was associated with marginal incremental value for both the prediction of histology and origin in the APS model. Overall, our study demonstrated a good predictive ability to determine patients’ histology and anatomical origin using [^18^F]F-FDG-PET/CT-derived radiomics features, mainly from CT.

## 1. Introduction

Cancer is considered a leading cause of death and a major factor of reduced life expectancy worldwide, reflected by the rapid increase in cancer incidence and mortality [1]. According to the World Health Organization (WHO), cancer ranks as the second most common cause of death before the age of 70 in the majority of countries [2]. The global cancer burden is expected to rise by almost 50% from 2020 to 2040, with a larger increase in emerging industrial countries due to demographic changes [3].

Medical imaging plays a fundamental role in cancer staging, and is needed in a substantial part of treatment decisions and prognostication nowadays. Standard radiological techniques, including innovative postprocessing tools, are expected to drive future risk stratification models, paying special attention to the field of radiomics. Radiomics refers to the extraction of high-dimensional data from various sources of state-of-the-art medical images using mathematical and machine learning methods, aiming to explore possible ties with biology, supporting clinical decision-making and overcoming the limitations of solely visual image interpretation [4,5]. Its general workflow includes image reconstruction, segmentation, feature extraction, feature selection, and data analysis [6]. Emerging data support the diagnostic and prognostic efficacy of radiomics in various malignancies, supporting the immense potential for computer-aided cancer detection, management, prognosis and surveillance in the future [7].

A recent study investigated the site-specific variation in radiomics features of head and neck squamous cell carcinoma [8], and several further studies demonstrated the ability of radiomics features to perform histological subtype classification among several cancer types [9,10,11]. In their limited to head and neck cancer, Liu et al. [8] thus suggested that the tumour site should be considered when developing radiomics-based models. Regarding subtype classification, Yang et al. [9] showed that their radiomics models on CT images could visualize significant differences between squamous cell carcinoma [SCC] and adenocarcinoma [AC], which could result in significantly different patient management. Their verification was performed on three independent databases, robustly suggesting the histology-specific radiomics values.

However, baseline knowledge about anatomical disease origin, and histology-specific radiomics variations across different malignancies, is lacking. Thus, the aim of our proof-of-principle study was to investigate whether radiomics features have the ability (1) to discriminate between histological subtypes (SCC vs. AC), (2) to predict the anatomical disease origin of different tumour entities (pulmonary vs. gastroesophageal vs. head and neck), and (3) to synchronously predict histology and anatomical disease origin, using baseline [^18^F]F-FDG PET/CT.

## 2. Methods

Based on an internal hospital database search engine (Montage, Montage Healthcare Solutions; accessed on March 2022), consecutive patients with either pulmonary, gastroesophageal or head and neck malignancies, who underwent [^18^F]F-FDG PET/CT as part of their initial staging between April 2015 and October 2020, were retrospectively identified. The tumours were either resected or biopsied as the standard of care for therapy planning and sent for histopathology evaluation.

Our study received institutional review board and local ethics committee approval (CAPCR/UHN REB #: 21-6113). Based on the retrospective nature of the study, the requirement for informed consent was waived by the ethics committee.

### 2.1. Image Acquisition

[^18^F]F-FDG PET/CT was acquired on a Siemens mCT40 PET/CT scanner (Siemens Healthineers, Erlangen, Germany) or on an in-line Biograph PET/CT scanner (Siemens Healthineers, Erlangen, Germany). Both scanners were from the same vendor, and acquisition and reconstruction parameters were harmonised (EARL-compliant) to minimise differences in image reconstruction and uptake values. Patients were imaged based on the standard of care protocol in our centre. They were positioned supine with images obtained from the top of the skull to the upper thighs. Patients were injected with 300–400 MBq (4–5 MBq/kg) of ^18^F-FDG after having fasted for 6 h, and PET/CT scanning was performed after 60 min. Overall, 5–9 bed positions were obtained, depending on patient height, with an acquisition time of 2–3 min per bed position. CT as part of PET/CT was performed using the following scan parameters: tube voltage 120 kV peak, 40–150 mAs, collimation 2 mm, rotation time 0.8 s, feed/rotation 8.4 mm. Images with a transverse pixel size of 1.00 and slice thickness of 5 mm were reconstructed in the axial plane using a soft tissue kernel. PET emission scan using time of flight with scatter correction was obtained covering the identical transverse field of view. PET parameters were as follows: pixel size was 2.6 mm × 2.6 mm, and slice width was 3.27 mm. A 4 mm full width at half maximum (FWHM) Gaussian filter type was used.

### 2.2. Image Segmentation and Radiomics Feature Extraction

Image segmentation and radiomics features extraction were performed with a commonly used open-source and IBSI-compliant software platform (LIFEx, Version 6.30; [12]) by one independent reader with 5 years of experience in oncologic radiology and hybrid imaging. Imaging segmentation and contouring have been described in detail before in other evaluations by our group [13]. The contours of the primary pulmonary, gastroesophageal and head and neck tumours were semi-automatically delineated and segmented using a threshold method, applying three different thresholds on the PET volumes of interest (VOI) (PET overall, PET SUVpeak, and PET 40% and 70% SUVmax thresholds) [13,14]. Notably, among them, the overall thresholding (contrast to the background) was the best method of thresholding, having the most significant variables in the further analyses. Volumetric segmentation of the primary tumours was performed manually on the low-dose unenhanced CT component (acquired as part of the [^18^F]F-FDG PET/CT) in a slice-by-slice fashion.

The extracted radiomics features included conventional metrics features, such as size and the voxel intensities’ mean, median, maximum, and minimum values. Additionally, shape histogram-based features were extracted, such as volume, compacity, and sphericity, including their asymmetry (skewness), flatness (kurtosis), uniformity and randomness, and textural features, i.e., neighbourhood grey-level different matrix (NGLDM), grey-level co-occurrence matrix (GLCM), grey-level zone length matrix (GLZLM), and grey-level run length matrix (GLRLM).

### 2.3. Prediction Models

Subsequently, different prediction models were built, in order to investigate whether [^18^F]F-FDG PET/CT–derived radiomics features have the ability (1) to discriminate between histological subtypes (SCC vs. AC), (2) to predict the anatomical disease origin of different tumour entities (pulmonary vs. gastroesophageal vs. head and neck), and (3) to synchronously predict histology and anatomical disease origin.

### 2.4. Statistical Analysis

One thousand cross-validation (CV) sets were created by randomly allocating 313 participants to a training set, and the remaining 78 participants to the corresponding testing set. Features missing on >20% of participants were removed, as were features with no variability. Notably, after excluding these features, no missing data remained in our database. To reduce the feature space prior to model building, data were normalized using a data-dependent algorithm [15], and then clustered using the complete clustering algorithm [16]. Fifteen clusters were created for each modality (CT, PET and PET/CT) based on our patient population. Based on five clusters (resulting in 15 variables because one from each group is selected), we were able to evaluate over 20 participants per variable in the training data. The goal of this step is to reduce the feature space to a more manageable level for further analysis. Overall, 20 participants per variable are generally considered enough to undertake multivariable regression.

The data-dependent normalization algorithm produces normalized data, useful for feature clustering, and while the resulting transformations can be stored, the likelihood of these transformations creating normalized data in an external validation sample is unknown. For this reason, we applied data-independent transformations to features with a mean/median ratio > 1.5 as follows: for features with positive values, only a log transformation was applied, and for features with negative or zero values, a square-root transformation was applied to the absolute values and the original sign retained. Modelling was performed on these data, separately for CT and PET features and for each of the 1000 CV training sets. Two machine learning modelling strategies were employed to predict histology (SCC vs. AC): all possible subset regression (APS) and recursive partitioning [17] (Figure 1).

Method 1 consisted of APS plus R^2^ for feature selection, and Method 2 was the combination of recursive partitioning and random forest selection. Hence, prior to APS modelling, one feature was selected from each of the 15 clusters by choosing the feature with the largest point biserial correlation, which also explained more than a five per cent variance in the outcome ($R^{2}>0.05$). This reduced the feature space to a maximum of 15 features, and logistic regression was performed with all possible feature subsets. The set of features which produced the highest Youden score (sensitivity + specificity − 1) was chosen as the final APS model for that CV set. The performance of this model was then evaluated on the corresponding testing set. Prior to recursive partitioning, one feature was selected from each cluster by choosing the feature with the smallest *p*-value that is also <0.1 on the Kruskal–Wallis test. Again, the feature space was reduced to a maximum of 15 features, and these were entered into a random forest model to further reduce the feature space to k features. The features chosen by the random forest model were then entered into k partitioning models, in order of importance so that the first model included only the most important features, the next model included the top two features, and so on. The maximum depth of the partitioning models was set to the number of features entered to prevent over-fitting. The model with the largest Kappa score was chosen as the final partitioning model and evaluated with the testing data. This process was completed separately for the CT and PET features, and then, for each CV training set, the modelling process was repeated with the final features from both the CT and PET models and evaluated on the testing set. Overall, the purpose was to try to discriminate patients, and not to conduct an assessment of the different methodologies, but to find a suitable model. For this, R^2^ is an established method of feature reduction, so was chosen for the regression. For the partitioning algorithm, features with the best chance of success with that algorithm were chosen, so features with the biggest between-group differences were selected. It is noteworthy that we accurately took care of the building model process to prevent any probable data leakage. All training was done only on the training set; we broke the training data into CV sets and then evaluated the models on the independent testing data.

To provide a benchmark for the model performance, a ‘Prevalence Only’ model was also fitted for each CV set. This model randomly allocated patients in the testing data to a group based only on trait prevalence in the training data. Statistical analyses were performed in the R programming language, version 4.2.1 [18].

## 3. Results

Overall, 391 consecutive patients (107 females, 284 males; mean age 66.7 ± 11.2 years; range: 22–92 years) with pulmonary (n = 142), gastroesophageal (n = 128) or head and neck (n = 121) malignancies, who underwent [^18^F]F-FDG PET/CT as part of their initial staging between April 2015 and October 2020 were included in our study. Our cohort comprised 219 SCCs and 172 ACs. Overall demographic patient data are provided in Table 1. 

### 3.1. Prediction of Histology

The recursive partitioning algorithm outperformed APS, and both methods were substantially better than the model based on prevalence only. In the APS model, the best predictive ability to determine histology was achieved when combining PET and CT radiomics features, resulting in a sensitivity of 0.78 and a specificity of 0.71, whereas, in the recursive partitioning algorithm, no incremental predictive value was attained when adding PET features to a model with CT radiomics features only. This model, using only CT radiomics features, showed a good predictive ability to determine histology (SCC vs. AC) with a sensitivity of 0.90 and specificity of 0.77 (Table 2 and Figure 2). The three most commonly selected CT radiomics features for both recursive partitioning algorithm and APS were GLZLM_ZLNU, GLRLM_LGRE and Conventional_HUQ1.

### 3.2. Prediction of Anatomical Disease Origin

A similar process was used to predict the anatomical origin of malignancy. However, only the recursive partitioning algorithm was used since this was a nominal three-level outcome (pulmonary vs. gastroesophageal vs. head and neck cancers). Similar to the prediction model for histology, the best-validated performance to determine anatomical disease origin was achieved using CT radiomics features only, resulting in overall good predictive ability for all cancer sites. In particular, pulmonary malignancies were identified with a high degree of accuracy (sensitivity 0.93; specificity 0.98). Overall results for all disease sites are displayed in Table 3 and Figure 3. The five most important CT radiomics features to predict disease origin were Discretized_HUQ2, GLZLM_SZLGR, GLZLM_ZLNU, Shape_Sphericity and Discretized_HUmin. 

### 3.3. Prediction of Histology and Anatomical Disease Origin

A model for the synchronous prediction of histology and anatomical disease origin resulted in high diagnostic accuracy in determining gastroesophageal AC (sensitivity 0.88; specificity 0.92), pulmonary AC (sensitivity 0.89; specificity 0.88) and head and neck SCC (sensitivity 0.91; specificity 0.92) using CT radiomics features (Table 4 and Figure 4). The five most important CT radiomics features for the combined model were Discretized_HUmin, Discretized_HUQ2, GLCM_Correlation, GLZLM_SZLGE and GLZLM_ZLNU.

## 4. Discussion

Our proof-of-principle study indicates that radiomics features, derived from standard of care [^18^F]F-FDG PET/CT, had the ability to determine histological subtypes (SCC vs. AC) and anatomical disease origin across different tumour entities (pulmonary vs. gastroesophageal vs. head and neck). However, it appears that the high statistical performance, at least in the partitioning model, was mainly derived from the CT component of the [^18^F]F-FDG PET/CT.

In the past decade, the field of radiomics has significantly evolved, enabling the extraction of high-dimensional data from different sources of medical images, including PET, and has shown promising results considering response and outcome prediction among various malignancies [19,20,21,22]. Several studies also investigated the ability of radiomics features to accurately classify histological cancer subtypes in a complementary, non-invasive manner. For example, studies by Hyun et al. [23] and Han et al. [24] used a PET/CT-based machine learning approach, which successfully discriminated AC from SCC in patients with non-small cell lung cancer (NSCLC). A further study by Du et al. [25] demonstrated the ability of multiphase CT radiomic features to differentiate AC from SCC in 260 patients with gastroesophageal cancer. The authors established a 3D-ROI arterial and venous phase-based model demonstrating slightly lower sensitivity in both models and in both training and validation cohorts as compared to our CT radiomics features model (partitioning algorithm) (0.872 vs. 0.90). Discriminating histological subtypes using a radiomics approach may be particularly important and useful, since radiomics features are usually derived from the entire tumour rather than just a biopsy sample, which might be susceptible to sampling errors [26]. Thus, radiomics data can provide important information regarding cancer histology and biology and can also be used for potential cross-validation.

However, to the best of our knowledge, no study so far investigated a radiomics-based prediction model using [^18^F]F-FDG PET/CT in order to classify histologic subtypes and discriminate anatomical disease origin across different malignancies. Thus, we wanted to assess our hypothetical assumption regarding the possibility of molecular-level discrimination of cancers regardless of their anatomical origin. This may guide us to a new paradigm to also look at malignancies from a molecular standpoint alongside the classic anatomical categorization. However, since we have only investigated limited types of cancers (pulmonary, gastroesophageal and head and neck malignancies), further investigations in broader spectrums are highly recommended.

The main findings of our study demonstrated a good predictive ability to differentiate between histological subtypes (AC vs. SCC) with a sensitivity of 0.90 and specificity of 0.77, and to determine the anatomical disease origin across patients with pulmonary (sensitivity of 0.93; specificity of 0.98), gastroesophageal (sensitivity of 0.90; specificity of 0.90) and head and neck (sensitivity of 0.83; specificity of 0.95) malignancies. For the discrimination of histological subtypes, two different methods were used in our study, APS and the partitioning algorithm. While APS fitted all possible combinations of independent variables, the recursive partitioning algorithm split the entire dataset to build a decision tree, striving to create increasingly homogenous sub-groups, thus inducing a partition on the space of explanatory variables. The advantage of the recursive partitioning algorithm was that it allowed varying prioritizing of misclassifications to create a decision rule with higher sensitivity or specificity, instead of modelling a linear relationship. Furthermore, the means (e.g., cross-validation) of fitting the data tree, helped to reduce overfitting, including unnecessary predictor variables, especially in cases with many potential predictors. On the other hand, the APS model gave more of an overview of all possible scenarios versus varying prioritization.

Finally, a model for the synchronous prediction of histology and anatomical disease origin resulted in high diagnostic accuracy to determine gastroesophageal AC (sensitivity of 0.88; specificity of 0.92), pulmonary AC (sensitivity of 0.89; specificity of 0.88) and head and neck SCC (sensitivity of 0.91; specificity of 0.92). Prediction of the latter, however, may certainly be related to the overall high incidence, and thus only represents limited clinical value. Only a moderate predictive ability was achieved for SCCs from pulmonary and gastroesophageal origin, which may be related to the lower prevalence of the respective diseases, reflected by rather small sample sizes in our cohort.

Furthermore, for all prediction models, mainly the CT radiomics features achieved the best validated statistical performance, and a combination with PET radiomics features was not associated with incremental predictive value. Similar findings were described by Lv et al. [27], comparing the prognostic ability of PET versus CT radiomics features among patients with nasopharyngeal cancer, achieving the best statistical performance using CT radiomic features only, which might be related to the higher resolution of CT images, as compared to PET.

Nevertheless, several studies so far have demonstrated the value of conventional PET radiomics features. For example, Salem et al. [28] and Lu et al. [29] demonstrated in patients with non-small cell lung cancer undergoing staging PET/CT, the ability of maximal standardized uptake values (SUVmax) to discriminate between AC and SCC. In this regard, some of the representative cases in our study population were provided in Figure 5, Figure 6 and Figure 7.

This emphasizes that PET may remain an important diagnostic tool, providing better disease visualization and revealing glucose metabolism, which is particularly beneficial in N- and M-staging. Thus, our study also adds to the literature that non-contrast CT radiomics derived from the CT component of the PET/CT may provide incremental information about pathophysiology and biology, further enhancing the diagnostic and predictive value of PET/CT, beyond staging and restaging purposes. Furthermore, the majority of prediction models pursue a multi-omics approach in combination with clinical parameters [13,30], whereas in our proof-of-principle study, the use of radiomics only already achieved a high degree of statistical performance, indicating further potential improvement of the predictive ability when adding clinical parameters.

The main strength of radiomics data in oncology is the fact that digital radiological images are obtained for almost every cancer patient [26]. Thus, radiomics analyses may contribute to the improvement of understanding tumour biology and the pursuit of precision medicine, in which such markers are used to determine optimal treatment and predict outcome and therapy response. Currently, the assessment of histology and molecular biomarkers, as well as mutational status, is based on invasive tissue sampling or surgical resection. Therefore, the usage of noninvasive imaging techniques for diagnosis and tumour characterization could represent a potentially complementary -omics data set or, eventually, even an alternative, especially in frail patients with advanced, inoperable disease or when inaccessible to biopsy.

Furthermore, additional baseline knowledge about site-specific radiomics variations may provide useful information in the assessment of complex diseases, in patients with secondary malignancies or in partly dedifferentiated tumors. Liu et al. [8] already demonstrated that radiomics features, derived from contrast-enhanced CT, are significantly dependent on the tumor location, assessed on a set of 605 consecutive patients with primary head and neck SCCs, arising from the oral cavity, oropharynx, larynx and hypopharynx. These findings are in line with the results of our study, indicating that the radiomics-based prediction models have the ability not only to predict anatomical tumour origin, but potentially being able to discriminate between secondary malignancies and tumour recurrence. 

The following study limitations must be acknowledged. First, there are inherent drawbacks due to the retrospective nature of the study. Second, we investigated a relatively inhomogeneous patient population, including different tumour stages/grades and, thus, we depicted different time points during the course of the disease. Also, there was an imbalance between Asian/non-Asian races, as well as in different sexes per diagnosis. Thus, further analysis in a more homogenous patient cohort is needed, to confirm and further enhance our results. Finally, the selection of the models used here is partly subjective. However, that is the case for several other studies using these methodologies as well. Ultimately, the purpose was to try to discriminate patients, not to conduct an assessment of the different methodologies.

## 5. Conclusions

Our proof-of-principle study may demonstrate a potentially high degree of diagnostic accuracy to predict histology and disease origin across different tumour entities, using the standard of care combined [^18^F]F-FDG PET/CT-derived radiomics features, mostly based on the CT component. This may help with the development of a new paradigm regarding the potential of also thinking about tumours from their molecular basis, and not only on the basis of their anatomical origin per se.

## Figures and Tables

**Figure 1 cancers-16-01873-f001:**
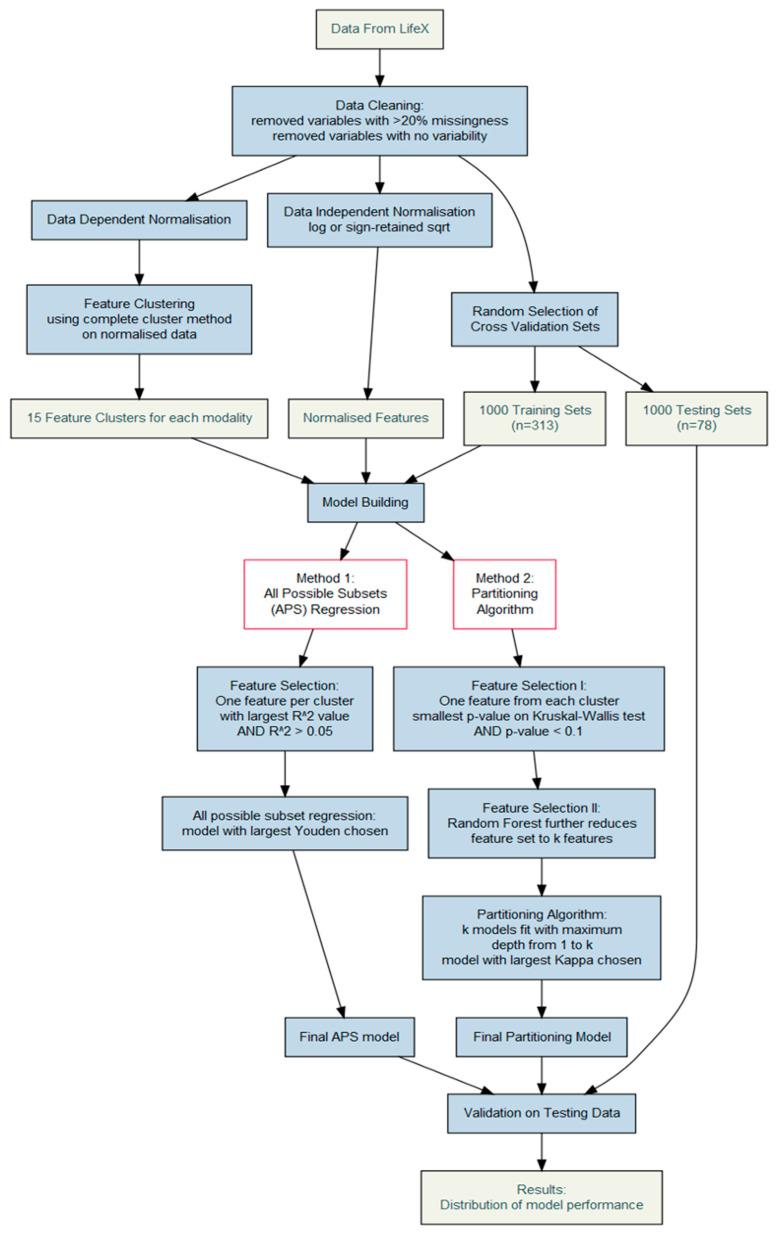
Illustration of the model building algorithms.

**Figure 2 cancers-16-01873-f002:**
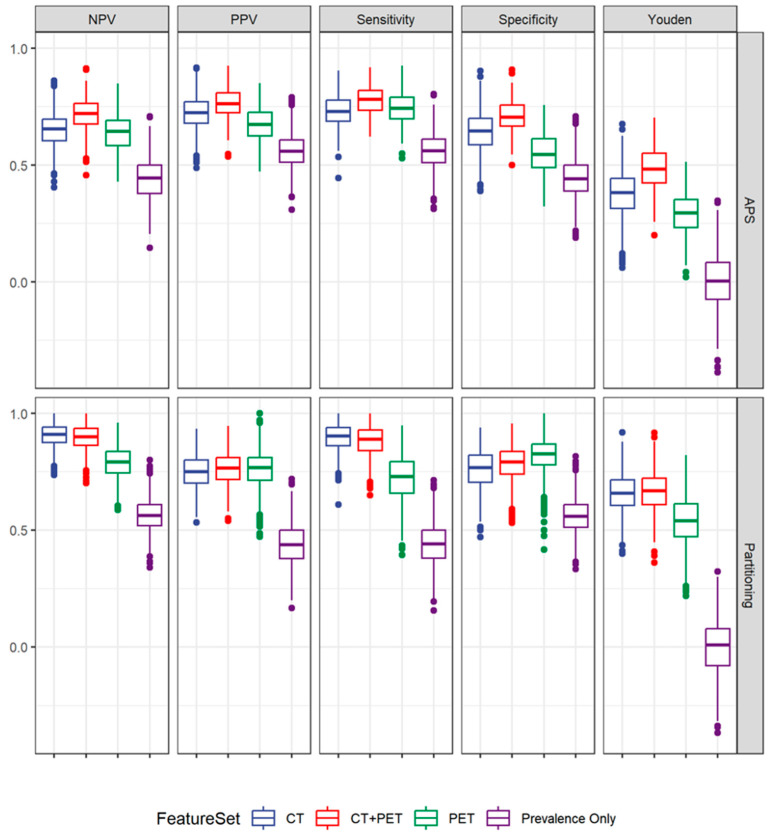
Distribution of performance metrics across the CV sets for CT, PET, PET/CT and a prevalence only model to predict histology. Results for all possible subsets (APS) regression are in the top row and partitioning algorithm results are in the bottom row. The recursive partitioning algorithm outperforms APS and both methods are substantially better than a model based on prevalence only.

**Figure 3 cancers-16-01873-f003:**
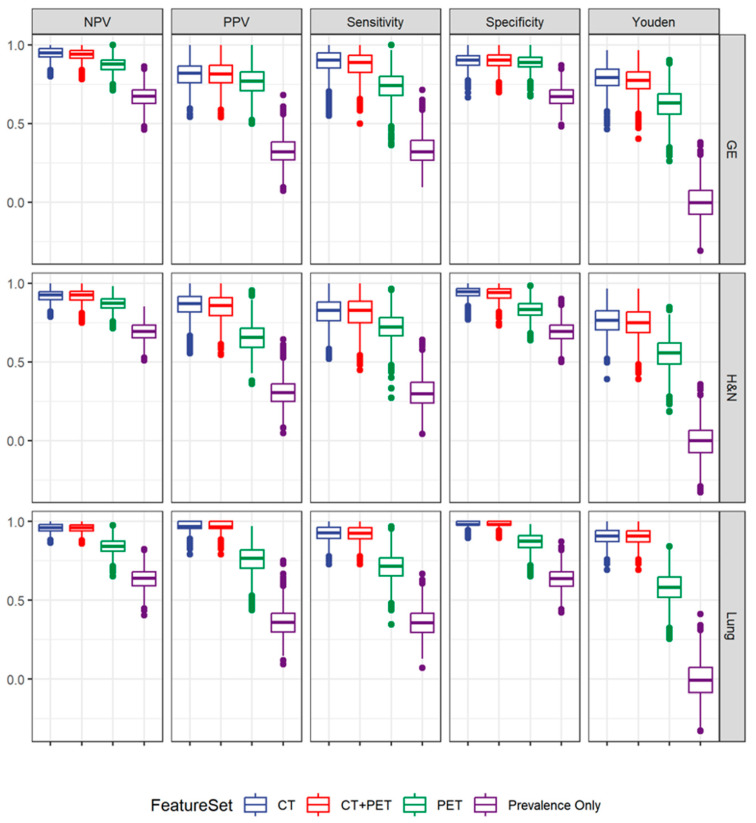
Class-specific performance statistics across modalities to predict anatomical disease origin. Both PET and CT radiomic features offer superior prediction to an uninformed model, which randomly predicts class based on prevalence alone.

**Figure 4 cancers-16-01873-f004:**
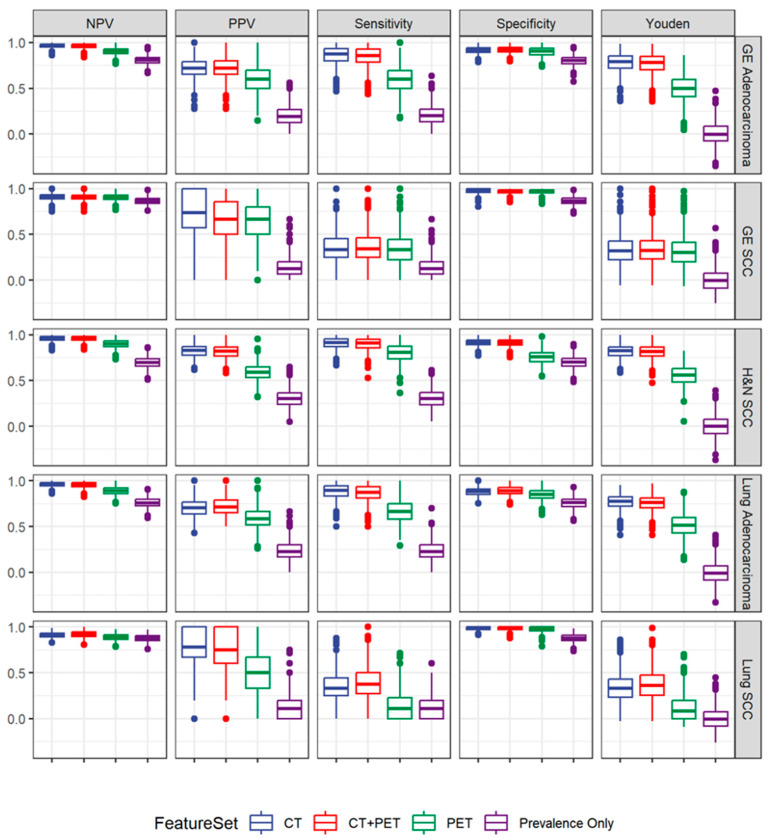
Class-specific performance statistics across modalities to synchronously predict anatomical disease origin and histology. The best statistical performance was achieved for the prediction of gastroesophageal and pulmonary AC and H&N SCC. Adding PET radiomics features to CT radiomics features only does not improve the predictive ability.

**Figure 5 cancers-16-01873-f005:**
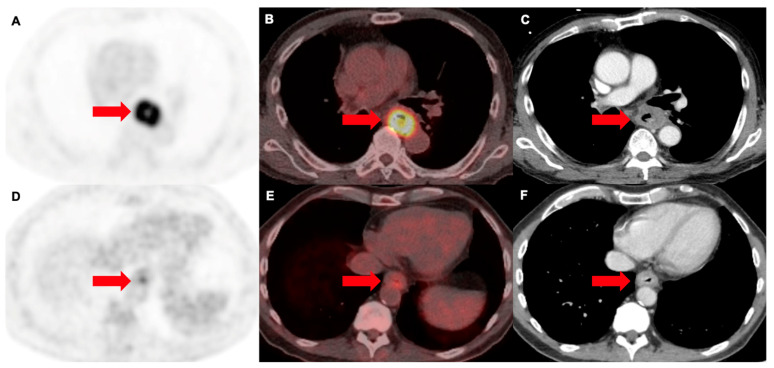
[^18^F]F-FDG PET/CT in two different patients with esophageal cancer (arrows). (**A**) PET, (**B**) PET/CT and (**C**) correlative contrast-enhanced CT from an 80 y.o. male patient, showing an intensely metabolically active circumferential mass in the distal oesophagus (SUVmax of 13.1), corresponding to a squamous cell carcinoma. (**D**) PET, (**E**) PET/CT and (**F**) correlative contrast-enhanced CT from a 67 y.o. male patient, with a circumferential mass in a similar location, however, with only mild FDG-uptake, corresponding to an adenocarcinoma (SUVmax of 3.4).

**Figure 6 cancers-16-01873-f006:**
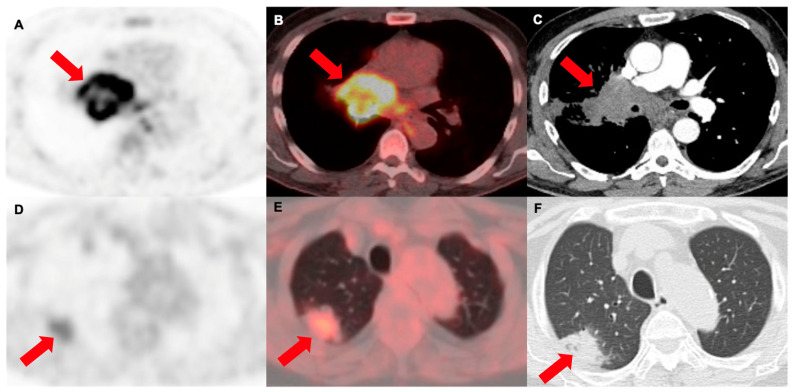
[^18^F]F-FDG PET/CT in two different patients with lung cancer (arrows). (**A**) PET, (**B**) PET/CT and (**C**) correlative contrast-enhanced CT from a 56 y.o. male patient with an intensely metabolically active, infiltrative central pulmonary mass in the right upper lobe (SUVmax of 10.9), corresponding to squamous cell carcinoma. (**D**) PET, (**E**) PET/CT and (**F**) correlative contrast-enhanced CT from a 71 y.o. female, showing a peripheral mass-like consolidation with an air bronchogram, with mildly increased [^18^F]F-FDG uptake, in keeping with an adenocarcinoma (SUVmax of 3.6).

**Figure 7 cancers-16-01873-f007:**
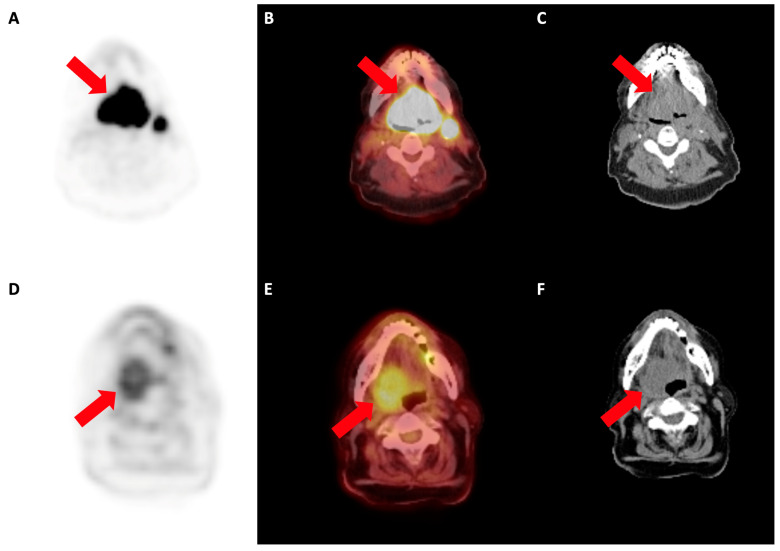
[^18^F]F-FDG PET/CT in two different patients with head and neck cancer (arrows). (**A**) PET, (**B**) PET/CT and (**C**) correlative CT from a 74 y.o. male patient with an intensely metabolically active, large mass centred within the base of the tongue (SUVmax of 40.1), corresponding to squamous cell carcinoma. (**D**) PET, (**E**) PET/CT and (**F**) correlative CT from an 80 y.o. female, showing a metabolically active mass in the right tongue base, with increased [^18^F]F-FDG uptake (not as much as the other patient), in keeping with an adenocarcinoma (SUVmax of 5.7).

**Table 1 cancers-16-01873-t001:** Study cohort characteristics.

Characteristics	n = 391
**Age**	
Mean (SD)	66.7 (11.2)
Median (Min, Max)	67 (22, 92)
**Sex**	
Female	107 (27%)
Male	284 (73%)
**Race**	
Asian	56 (14%)
Non-Asian	335 (86%)
**BMI**	26.2 (5.6)
**History of Smoking**	259 (66.2%)
**History of Regular Alcohol Intake**	174 (44.5%)
**Malignancy**	
Gastroesophageal	128 (33%)
Head and Neck	121 (31%)
Pulmonary	142 (36%)
**Histology**	
** *Adenocarcinoma* **	172 (44%)
Gastroesophageal	76 (44%)
Head and Neck	2 (1%)
Pulmonary	94 (55%)
** *Squamous cell carcinoma* **	219 (56%)
Gastroesophageal	52 (24%)
Head and Neck	119 (54%)
Pulmonary	48 (22%)

**Table 2 cancers-16-01873-t002:** The median and the values corresponding to the central 95% of validation statistics were evaluated across 1000 CV sets to predict histology using [^18^F]F-FDG PET/CT radiomics features. Not that, for a partitioning algorithm, there are multiple cut-points. Consequently, ROC curve analysis was not performed and, hence, there were no AUC statistics.

	Prevalence	CT	PET	PET/CT
**APS**				
AUC	0.53 (0.43, 0.63)	0.69 (0.59, 0.78)	0.65 (0.56, 0.74)	0.74 (0.65, 0.82)
NPV	0.44 (0.29, 0.59)	0.66 (0.50, 0.81)	0.65 (0.47, 0.81)	0.72 (0.56, 0.86)
PPV	0.56 (0.41, 0.71)	0.73 (0.60, 0.85)	0.67 (0.53, 0.80)	0.76 (0.64, 0.88)
Sensitivity	0.56 (0.41, 0.71)	0.73 (0.60, 0.86)	0.74 (0.61, 0.88)	0.78 (0.64, 0.90)
Specificity	0.44 (0.28, 0.62)	0.65 (0.48, 0.79)	0.55 (0.38, 0.72)	0.71 (0.56, 0.85)
Youden Index	0.00 (−0.22, 0.22)	0.38 (0.18, 0.56)	0.30 (0.11, 0.48)	0.48 (0.29, 0.65)
**Partitioning**				
NPV	0.56 (0.42, 0.70)	0.91 (0.80, 1.00)	0.79 (0.66, 0.91)	0.90 (0.79, 0.98)
PPV	0.44 (0.29, 0.61)	0.75 (0.61, 0.88)	0.77 (0.59, 0.90)	0.77 (0.62, 0.90)
Sensitivity	0.44 (0.29, 0.62)	0.90 (0.77, 1.00)	0.73 (0.54, 0.88)	0.89 (0.74, 0.97)
Specificity	0.56 (0.42, 0.71)	0.77 (0.60, 0.90)	0.83 (0.65, 0.93)	0.79 (0.61, 0.91)
Youden Index	0.01 (−0.21, 0.23)	0.66 (0.50, 0.80)	0.54 (0.31, 0.73)	0.67 (0.50, 0.82)

APS: all possible subset regression.

**Table 3 cancers-16-01873-t003:** The median and the values corresponding to the central 95% of validation statistics were evaluated across 1000 CV sets to predict anatomical disease origin using [^18^F]F-FDG PET/CT radiomics features.

	Prevalence	CT	PET	PET/CT
**GE**				
NPV	0.67 (0.55, 0.79)	0.95 (0.85, 1.00)	0.88 (0.78, 0.96)	0.94 (0.84, 1.00)
PPV	0.32 (0.17, 0.52)	0.82 (0.66, 0.95)	0.77 (0.59, 0.93)	0.81 (0.64, 0.96)
Sensitivity	0.32 (0.16, 0.52)	0.90 (0.71, 1.00)	0.74 (0.54, 0.91)	0.89 (0.68, 1.00)
Specificity	0.67 (0.54, 0.80)	0.90 (0.79, 0.98)	0.89 (0.79, 0.98)	0.90 (0.76, 0.98)
Youden	0.00 (−0.22, 0.24)	0.79 (0.61, 0.92)	0.63 (0.42, 0.81)	0.78 (0.58, 0.92)
**H&N**				
NPV	0.69 (0.57, 0.80)	0.93 (0.84, 0.98)	0.87 (0.78, 0.96)	0.92 (0.82, 0.98)
PPV	0.30 (0.14, 0.50)	0.87 (0.68, 1.00)	0.66 (0.48, 0.85)	0.86 (0.65, 1.00)
Sensitivity	0.30 (0.12, 0.48)	0.83 (0.62, 0.96)	0.72 (0.52, 0.90)	0.83 (0.57, 0.96)
Specificity	0.69 (0.56, 0.82)	0.95 (0.85, 1.00)	0.83 (0.71, 0.93)	0.94 (0.82, 1.00)
Youden	0.00 (−0.23, 0.22)	0.76 (0.56, 0.91)	0.56 (0.34, 0.73)	0.75 (0.54, 0.91)
**Pulmonary**				
NPV	0.64 (0.51, 0.76)	0.96 (0.90, 1.00)	0.84 (0.75, 0.94)	0.96 (0.89, 1.00)
PPV	0.36 (0.19, 0.55)	0.97 (0.89, 1.00)	0.77 (0.58, 0.91)	0.97 (0.89, 1.00)
Sensitivity	0.36 (0.19, 0.54)	0.93 (0.82, 1.00)	0.71 (0.54, 0.88)	0.93 (0.81, 1.00)
Specificity	0.64 (0.49, 0.78)	0.98 (0.94, 1.00)	0.88 (0.73, 0.96)	0.98 (0.94, 1.00)
Youden	−0.01 (−0.23, 0.24)	0.91 (0.80, 1.00)	0.58 (0.38, 0.76)	0.91 (0.80, 1.00)

GE: gastroesophageal; H&N: head and neck.

**Table 4 cancers-16-01873-t004:** The median and the values corresponding to the central 95% of validation statistics were evaluated across 1000 CV sets to predict disease origin and histology using [^18^F]F-FDG PET/CT radiomics features.

	Prevalence	CT	PET	PET/CT
**GE AC**				
NPV	0.81 (0.71, 0.90)	0.97 (0.91, 1.00)	0.91 (0.83, 0.97)	0.97 (0.90, 1.00)
PPV	0.19 (0.00, 0.42)	0.72 (0.50, 0.92)	0.60 (0.33, 0.88)	0.72 (0.50, 0.92)
Sensitivity	0.20 (0.00, 0.44)	0.88 (0.65, 1.00)	0.60 (0.31, 0.85)	0.86 (0.62, 1.00)
Specificity	0.80 (0.69, 0.89)	0.92 (0.84, 0.98)	0.91 (0.79, 0.98)	0.92 (0.84, 0.98)
Youden	0.00 (−0.21, 0.25)	0.79 (0.56, 0.95)	0.50 (0.21, 0.75)	0.78 (0.53, 0.95)
**GE SCC**				
NPV	0.87 (0.79, 0.94)	0.91 (0.84, 0.97)	0.91 (0.83, 0.97)	0.91 (0.84, 0.97)
PPV	0.12 (0.00, 0.40)	0.74 (0.25, 1.00)	0.67 (0.20, 1.00)	0.67 (0.20, 1.00)
Sensitivity	0.12 (0.00, 0.40)	0.33 (0.08, 0.67)	0.33 (0.06, 0.67)	0.34 (0.08, 0.70)
Specificity	0.87 (0.77, 0.94)	0.98 (0.92, 1.00)	0.97 (0.91, 1.00)	0.97 (0.91, 1.00)
Youden	0.00 (−0.18, 0.27)	0.32 (0.04, 0.64)	0.30 (0.00, 0.64)	0.33 (0.04, 0.66)
**H&N SCC**				
NPV	0.70 (0.58, 0.81)	0.96 (0.89, 1.00)	0.90 (0.80, 0.98)	0.96 (0.89, 1.00)
PPV	0.30 (0.14, 0.48)	0.83 (0.67, 0.95)	0.59 (0.43, 0.77)	0.82 (0.66, 0.94)
Sensitivity	0.30 (0.14, 0.50)	0.91 (0.76, 1.00)	0.81 (0.62, 0.96)	0.91 (0.76, 1.00)
Specificity	0.70 (0.57, 0.81)	0.92 (0.84, 0.98)	0.76 (0.61, 0.89)	0.91 (0.83, 0.98)
Youden	0.00 (−0.21, 0.22)	0.82 (0.67, 0.94)	0.56 (0.35, 0.73)	0.82 (0.66, 0.94)
**Pulmonary AC**				
NPV	0.76 (0.65, 0.86)	0.96 (0.90, 1.00)	0.89 (0.80, 0.97)	0.96 (0.89, 1.00)
PPV	0.23 (0.06, 0.45)	0.70 (0.52, 0.88)	0.58 (0.37, 0.79)	0.71 (0.54, 0.90)
Sensitivity	0.23 (0.06, 0.45)	0.89 (0.73, 1.00)	0.67 (0.41, 0.89)	0.88 (0.67, 1.00)
Specificity	0.76 (0.65, 0.86)	0.88 (0.80, 0.96)	0.85 (0.73, 0.95)	0.89 (0.80, 0.97)
Youden	−0.01 (−0.22, 0.23)	0.77 (0.62, 0.90)	0.51 (0.27, 0.75)	0.76 (0.59, 0.90)
**Pulmonary SCC**				
NPV	0.88 (0.81, 0.94)	0.92 (0.85, 0.97)	0.89 (0.82, 0.95)	0.92 (0.85, 0.97)
PPV	0.11 (0.00, 0.40)	0.78 (0.33, 1.00)	0.50 (0.00, 1.00)	0.75 (0.33, 1.00)
Sensitivity	0.11 (0.00, 0.36)	0.33 (0.00, 0.67)	0.11 (0.00, 0.50)	0.38 (0.09, 0.75)
Specificity	0.88 (0.79, 0.95)	0.99 (0.95, 1.00)	0.99 (0.91, 1.00)	0.98 (0.93, 1.00)
Youden	−0.01 (−0.18, 0.25)	0.33 (0.00, 0.65)	0.09 (−0.04, 0.42)	0.36 (0.07, 0.72)

GE: gastroesophageal; H&N: head and neck; AC: adenocarcinoma; SCC: squamous cell carcinoma.

## Data Availability

The datasets used and/or analysed during the current study are available from the corresponding author on reasonable request.

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
