# Peer review of "The [18F]F-FDG PET/CT Radiomics Classifier of Histologic Subtypes and Anatomical Disease Origins across Various Malignancies: A Proof-of-Principle Study"

_cancers, 2024, doi:10.3390/cancers16101873_

Round 1
Reviewer 1 Report
Comments and Suggestions for Authors
This paper presents a model for predicting anatomical disease origin and histological subtype from PET/CT images.
My main concern is that in a radiomics study of primary tumors, the anatomical origin of the disease is always known a priori, because the lesion has to be segmented and the location of the lesion has to be identified beforehand. From a technical point of view, it is possible to use radiomics to predict tumor location. However, from a practical point of view, radiomics is expected to answer questions that may relate to patient management (e.g. predicting patient response or survival) or to a better understanding of a disease (e.g. distinguishing between different molecular subtypes from phenotypic data). Therefore, I see a real interest in identifying histological subtypes. I would suggest to the authors to redefine the questions of the predictive model or at least to justify any clinical interest to perform this investigation.
I include below some other minor comments:
ï‚· “PET parameters were as follows: image size: 2.6 pixels; slice: 3.27; and 4-mm full 104 width at half maximum (FWHM) Gaussian filter type.” ïƒ Please, use mm as units: Pixel size was 2.6 mm x 2.6 mm and slice width was 3.27 mm.
ï‚· Authors have use three different thresholds for lesion segmentation. Which of them have they selected for subsequent radiomics extraction?
ï‚· Could the author specify the CT reconstruction algorithm? Which kernel have they used?
ï‚· For CT and PET, have the authors used the same list of features?
ï‚· How was the histology confirmed? Have all patients undergone surgery? Please include this information in the methods section.
Author Response
Distinguished Reviewer #1
This paper presents a model for predicting anatomical disease origin and histological subtype from PET/CT images.
My main concern is that in a radiomics study of primary tumors, the anatomical origin of the disease is always known a priori, because the lesion has to be segmented and the location of the lesion has to be identified beforehand. From a technical point of view, it is possible to use radiomics to predict tumor location. However, from a practical point of view, radiomics is expected to answer questions that may relate to patient management (e.g. predicting patient response or survival) or to a better understanding of a disease (e.g. distinguishing between different molecular subtypes from phenotypic data). Therefore, I see a real interest in identifying histological subtypes. I would suggest to the authors to redefine the questions of the predictive model or at least to justify any clinical interest to perform this investigation.
A: Thanks for this comment. The reviewer is certainly right in regard to the practical component, the lesion needs to be identified beforehand. However, the identification is done by the reader who is doing the contouring, not by any system/algorithm itself. Thus, the generated radiomics data is indifferent to the actual location. The reviewer is also correct in that several radiomic studies are focusing on the evaluation of response or survival prediction. However, there is currently a discussion about if tumours should not be classified, staged and treated based on their anatomical location but based on their molecular profile. This is the reason why we wanted to evaluate in a proof-of-principle study if radiomics would be able to differentiate these tumour types, irrespective of their location.
I include below some other minor comments:
- “PET parameters were as follows: image size: 2.6 pixels; slice: 3.27; and 4-mm full 104 width at half maximum (FWHM) Gaussian filter type.” à Please, use mm as units: Pixel size was 2.6 mm x 2.6 mm and slice width was 3.27 mm.
A: Thanks for the comment. We revised the part accordingly.
- Authors have use three different thresholds for lesion segmentation. Which of them have they selected for subsequent radiomics extraction?
A: Thanks for this critical point. The thresholds were only for PET segmentation (not applicable to CT feature extraction). We used the output of all thresholds for modelling. However, as the initial step in data preprocessing, we evaluated the variables and only selected the best parameters out of the highly collinear factors. Thus, for final modeling we included the results of the overall thresholding since this method provided more statistically significant features when compared to the others. This is now added to MM.
- Could the author specify the CT reconstruction algorithm? Which kernel have they used?
A: Thanks for the comment. We added the details.
- For CT and PET, have the authors used the same list of features?
A: We used the well-established parameters in the field of radiomics provided by the dedicated software for this purpose (LIFEx). Some features were PET-specific (such as SUVs, TLG and MTV), and some were CT-specific (e.g., Hu-related parameters). Thus, while several feature categories do overlap, not all of the features are and cannot be the same.
- How was the histology confirmed? Have all patients undergone surgery? Please include this information in the methods section.
A: Many thanks for your critical comment. Yes, the tumours were either resected or biopsied as standard of care for therapy planning and sent for histopathology evaluation. We added this to the methods.

Reviewer 2 Report
Comments and Suggestions for Authors
This image study is important to improve diagnosis of certain tumors/cancers.
Comments:
1. The introduction can be a little bit in details and thus references too.
2. On Table 1: please explain why race is divided by Asian vs non-Asian?
3. On Table 1: please explain why 3 specific malignancy category?
4. ON Table 1: Please add information of the patients' BMI, smoking and alcohol history?
5. On Table 1: any specific cancer/tumor markers?
6. Please show head/neck samples similar to Figure 5 and figure 6. Similar to two different patients with head/neck caner.
Author Response
Distinguished Reviewer #2
This image study is important to improve diagnosis of certain tumors/cancers.
- The introduction can be a little bit in details and thus references too.
A: Thanks for your comment. We expanded the introduction based on your suggestion.
- On Table 1: please explain why race is divided by Asian vs non-Asian?
A: Many publications now require details about race/ethnicity. While we have a very diverse population visiting our centre/being referred here, not all of this information is accessible in our hospital information system. This is the only difference we reliably could report, we do not have information about white/non-white, black etc.. No other statistical evaluation is based on this. If the reviewer / editor feels it’s not necessary to report, we can take it out.
- On Table 1: please explain why 3 specific malignancy category?
A: We used these three malignancies since these are common malignancies in our referral population in PET/CT. Not all indications which are internationally standard in PET/CT are reimbursed in our health care system so we would see lower numbers. Also, we aimed for a balanced distribution of adenocarcinoma vs. squamous cell carcinoma in the entire population and those three malignancies lend themselves to those criteria.
- ON Table 1: Please add information of the patients' BMI, smoking and alcohol history?
A: Thanks for your comment. We completely agree that adding these data would help better representation of our study population. They were added to Table 1.
- On Table 1: any specific cancer/tumor markers?
A: Thanks for this comment. We would generally include certain tumours markers if we i.e. aiming for building a prognostic model of tumours marker together with distinct radiomic parameters. However, this was not the intention for this proof-of-principle study and they are therefore not included.
- Please show head/neck samples similar to Figure 5 and figure 6. Similar to two different patients with head/neck caner.
A: Thanks for your suggestion. A new figure was added (Figure 7).

Reviewer 3 Report
Comments and Suggestions for Authors
The authors present an interesting study in which they compare various imaging techniques in their ability to characterise and profile different physiological malignancies in the body. With access to nearly 400 participants, within which malignancies presenting in the pulmonary, gastroesophageal, or head and neck regions were known, the authors employ their technique and demonstrate good predictive value in such for determining indices such as anatomical origin and histology. Overall, this was a very well put together study that is comprehensive in both its approach and also writing, and in reviewing the manuscript I have only minor suggestions.
1. The formatting, in particular the alignment of text, within the tables, could be improved upon.
2. In images 5 and 6, arrows which point to areas of interest may be of benefit to the reader.
Author Response
Distinguished Reviewer #3
The authors present an interesting study in which they compare various imaging techniques in their ability to characterise and profile different physiological malignancies in the body. With access to nearly 400 participants, within which malignancies presenting in the pulmonary, gastroesophageal, or head and neck regions were known, the authors employ their technique and demonstrate good predictive value in such for determining indices such as anatomical origin and histology. Overall, this was a very well put together study that is comprehensive in both its approach and also writing, and in reviewing the manuscript I have only minor suggestions.
- The formatting, in particular the alignment of text, within the tables, could be improved upon.
A: Thanks for the point. It was the prepared draft by the editorial office. We will for sure again check the resubmission to ensure convenience for readers.
- In images 5 and 6, arrows which point to areas of interest may be of benefit to the reader.
A: Thanks for your suggestion. We added the arrows.

Round 2
Reviewer 1 Report
Comments and Suggestions for Authors
In the revised submission, the authors have properly addressed my comments. I have no further comments.
Reviewer 2 Report
Comments and Suggestions for Authors
The authors answered my comments. No more comments.